🔓 | **Open Peer Review** | Plant Microbiology | Observation

# The timing of bacterial mesophyll infection shapes the leaf chemical landscape

Veronica Roman-Reyna,[1,2] Nathaniel Heiden,[1,2] Jules Butchacas,[1,2] Hannah Toth,[1,2] Jessica L. Cooperstone,[3,4] Jonathan M. Jacobs[1,2]

**ABSTRACT** Chemistry in eukaryotic intercellular spaces is shaped by both hosts and symbiotic microorganisms such as bacteria. Pathogenic microorganisms like barley-associated *Xanthomonas translucens* (Xt) swiftly overtake the inner leaf tissue becoming the dominant microbial community member during disease development. The dynamic metabolic changes due to Xt pathogenesis in the mesophyll spaces remain unknown. Genomic group I of Xt consists of two barley-infecting lineages: pathovar translucens (Xtt) and pathovar undulosa (Xtu). Xtu and Xtt, although genomically distinct, cause similar water-soaked lesions. To define the metabolic signals associated with inner leaf colonization, we used untargeted metabolomics to characterize Xtu and Xtt metabolism signatures associated with mesophyll growth. We found that mesophyll apoplast fluid from infected tissue yielded a distinct metabolic profile and shift from catabolic to anabolic processes over time compared to water-infiltrated control. The pathways with the most differentially expressed metabolites by time were glycolysis, tricarboxylic acid cycle, sucrose metabolism, pentose interconversion, amino acids, galactose, and purine metabolism. Hierarchical clustering and principal component analysis showed that metabolic changes were more affected by the time point rather than the individual colonization of the inner leaves by Xtt compared to Xtu. Overall, in this study, we identified metabolic pathways that explain carbon and nitrogen usage during host–bacterial interactions over time for mesophyll tissue colonization. This foundational research provides initial insights into shared metabolic strategies of inner leaf colonization niche occupation by related but phylogenetically distinct phyllosphere bacteria.

**IMPORTANCE** The phyllosphere is a habitat for microorganisms including pathogenic bacteria. Metabolic shifts in the inner leaf spaces for most plant–microbe interactions are unknown, especially for *Xanthomonas* species in understudied plants like barley (*Hordeum vulgare*). *Xanthomonas translucens* pv. translucens (Xtt) and *Xanthomonas translucens* pv. undulosa (Xtu) are phylogenomically distinct, but both colonize barley leaves for pathogenesis. In this study, we used untargeted metabolomics to shed light on Xtu and Xtt metabolic signatures. Our findings revealed a dynamic metabolic landscape that changes over time, rather than exhibiting a pattern associated with individual pathovars. These results provide initial insights into the metabolic mechanisms of *X. translucens* inner leaf pathogenesis.

**KEYWORDS** *Xanthomonas translucens*, barley mesophyll, primary metabolism, untargeted metabolomics

The phyllosphere is a habitat for microorganisms including pathogenic bacteria. *Xanthomonas* species colonize the apoplast spaces and often displace most of the microbial community. It is hypothesized that *Xanthomonas* pathogens manipulate the host environment to acquire the metabolites for multiplication, and in return, plants restrict their colonization by regulating nutrient distribution (1, 2). *Xanthomonas*

Address correspondence to Jonathan M. Jacobs, jacobs.1080@osu.edu.

The authors declare no conflict of interest.

See the funding table on p. 6.

*translucens* (Xt) are mesophyll apoplast-infecting pathogens that colonize the intercellular foliar tissues. Xt includes two distinct barley pathogenic lineages: Xt pathovars translucens (Xtt), which is specific to barley, and undulosa (Xtu), which has a broader host range. Although evolutionarily divergent, both cause similar water-soaking symptoms in leaves (3, 4). For both, the colonization effect on the metabolic landscape on the leaf environment remains undescribed. To understand the metabolic landscape during Xt infection, we profiled barley mesophyll apoplastic fluids (BMAFs). We extracted BMAF from water-infiltrated and Xt-infected (Xtt UPB886 and Xtu UPB513) barley leaves at 6, 12, and 24 hours post-inoculation (hpi) (5–7). The BMAFs were sent to the West Coast Metabolic Center (UC Davis, California) for the screening of primary metabolites by gas chromatography time of flight mass spectrometry. The data were analyzed using MetaboAnalyst version 5.0 (8).

## Temporal dynamics during pathovar infections

The most dramatic differences in the metabolic profiles happened at 24 hpi (Fig. 1A), and the analyses detected 89 metabolites with significant log2 fold changes (FCs) compared to the water-infiltrated control across all conditions [paired *t*-test, false discovery rate (FDR) <0.05] (Fig. 1B). To avoid effects associated with infiltration procedures, water-inoculated samples were kept for each time point for fold change comparisons. Clustering revealed that time influenced the chemical changes during Xt inner leaf colonization more than pathogen lineage. From the total significantly enriched metabolites at each time point, 6 hpi had more than 60% of metabolites metabolized compared to 6 hours of water-inoculated samples (FC < 0,), while 12 and 24 hpi had 80%–100% of metabolites synthesized compared to 12 and 24 hours, respectively, water-inoculated samples (FC > 0) (Fig. 1B). Xtu- and Xtt-BMAF had similar metabolic profiles over time, but Xtu-treated displayed a broader array of metabolites.

## Nutrient relocation patterns during pathovar infections

We classified the metabolites into amines, amino acids, organic acids, sugars, and nucleosides (Fig. 1B). Some nucleosides changed during infection, while the above groups showed varying trends. Amines, nutrient sources for bacteria, are associated with reactive oxygen species responses (9, 10). Putrescine increased in Xtu-BMAF (12 hpi), while spermidine accumulated in Xtt-BMAF (24 hpi). Xt could also decarboxylate putrescine and spermidine from arginine or glutamine. Amino acids, nitrogen sources for both plants and bacteria, varied throughout the infection. At 6 hpi, amino acids were depleted in Xtu-BMAF and accumulated in Xtt-BMAF. At 12 and 24 hpi, Xtu and Xtt only shared glutamine and aspartate trends. The metabolite 5′-methylthioadenosine (MTA) is involved in recycling sulfur-containing amino acids and is a product of polyamine synthesis (11). MTA accumulated in both pathovar-infected samples at 24 hours, possibly as a response to changes in amino acids and amines. Organic acids (OAs) and sugars were consumed at 6 hpi and were more abundant in Xtu-BMAF, probably related to higher anabolic processes associated with stress responses. At 12 hpi, Xtu treatment accumulated nine OAs while Xtt treatment had only one. At 24 hpi, citric acid was the most abundant OA in both pathovars. Citric acid or citrate is part of the tricarboxylic acid cycle (TCA) and regulates glycolysis and fatty acid synthesis (12). Sugars accumulated at 12 and 24 hpi in Xtu- and Xtt-BMAF, with Xtu-treated leaves having twice as many sugars as Xtt treatment (compared to the water-infiltrated control). Cellobiose, glucose-6-phosphate, and sucrose had the highest FC. Sucrose, an important molecule in both plant and bacterial metabolism, exhibited different patterns (13–16). In Xtt treatment, sucrose decreased (6 and 24 hpi), while in Xtu treatment, it accumulated (12 hpi). Xtt likely uses sucrose to metabolize other sugars since glucose and fructose were not abundant or were immediately consumed. In Xtu-infected samples, sucrose may be broken down into fructose and glucose, as was found for other *Xanthomonas* (13). Overall, nutrient relocation patterns indicated distinct metabolite utilization by each

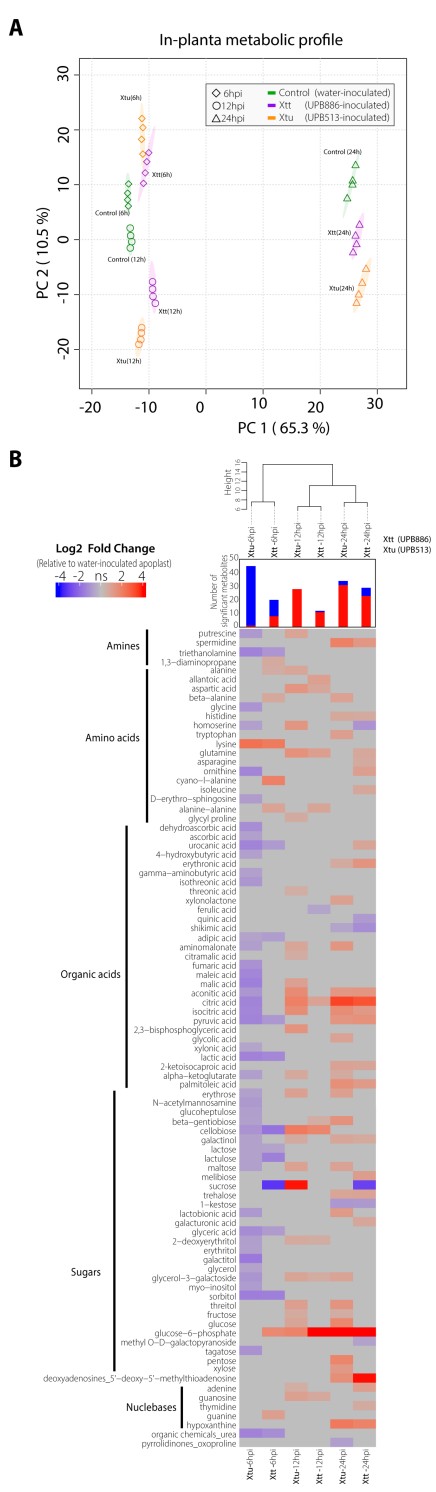

**FIG 1** Untargeted metabolic profile of *in planta Xt* growth. (A) Principal component analysis for each condition. Control, apoplast from leaves inoculated with water indicated as green. Xtu, apoplast from leaves inoculated with Xtu-UPB513 indicated as orange. Xtt, apoplast from leaves inoculated with Xtt-UPB886 indicated as purple. Each time point has four biological replicates. (B) Metabolic profile of Xtu- and Xtt-inoculated samples. The log2 FC for each infected sample is compared to water-infiltrated samples (control) at each time point. The hierarchical cluster used the Ward.D clustering algorithm and Euclidean distance method. The histogram indicates the number of significant metabolites based on log2 FC. The heatmap shows metabolites with FC ± 1.5 and a FDR < 0.05. Gray shows metabolites with no significant (ns) differences. Blue indicates FC > 0, and red indicates FC < 0.

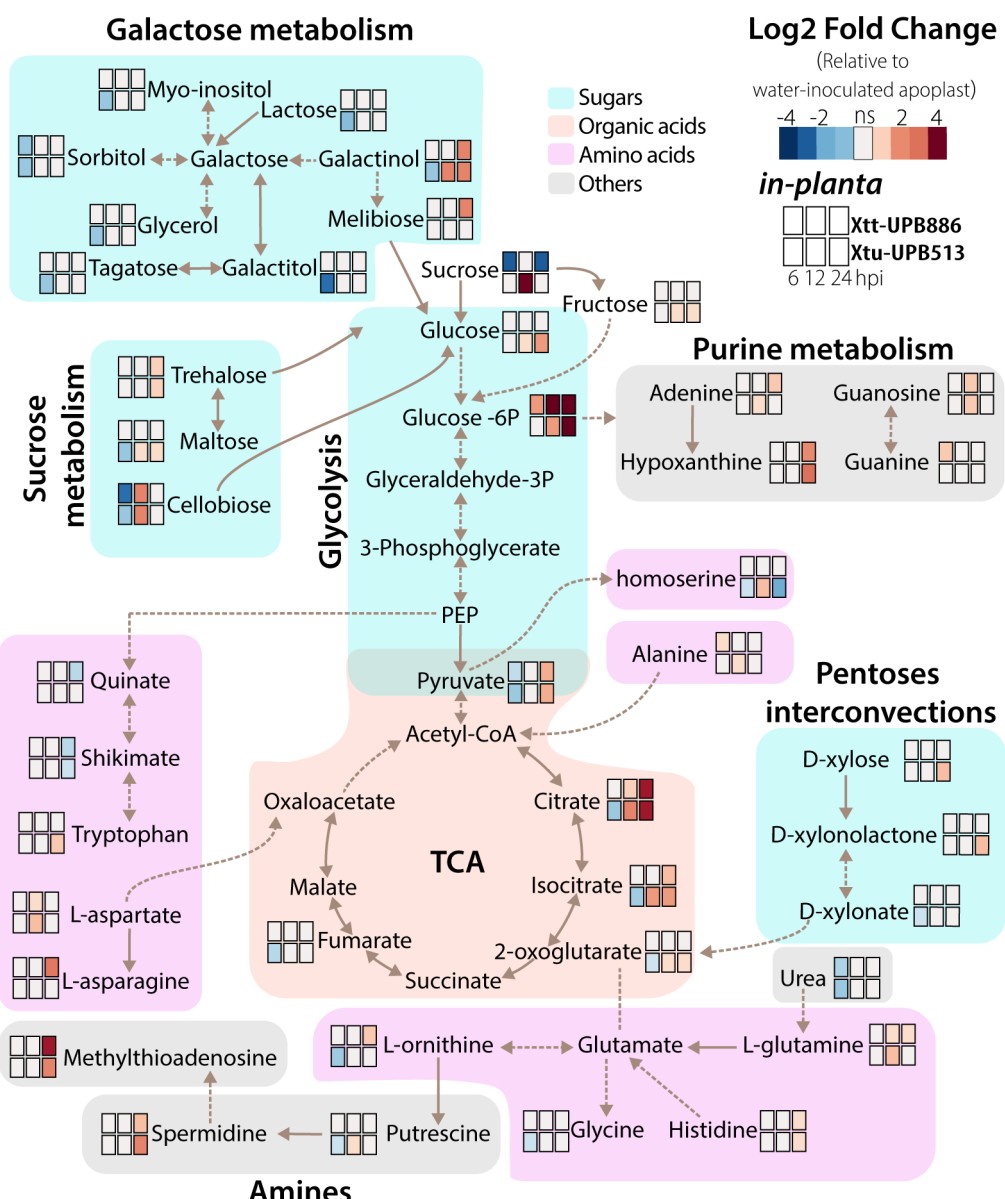

**FIG 2** Carbon and nitrogen KEGG-based metabolic pathways for *in planta* significant metabolites. Values are based on the heatmap from Fig. 1. Blue rectangles indicate FC > 0, red indicates FC < 0, and light gray shows ns differences. Fold changes are compared to water-inoculated control.

pathovar during early infection (6 hpi), with conserved host responses or bacterial nutrient requirements as the disease progressed.

## Metabolic pathway differences during pathovar infections

We identified pathways that had a minimum of two metabolites: amino acid metabolism, sugar metabolism (glycolysis, sucrose, galactose metabolism, and pentose interconversion), organic acid metabolism (TCA), and amine and purine metabolism (Fig. 2) (17). Glycolysis and TCA metabolic processes are important components of cellular respiration. Metabolites produced by these processes exhibit distinctive patterns, with Xtt-BMAF metabolites accumulated at 24 hpi, while Xtu-BMAF showed a decrease (6 hpi) followed by an increase (12 and 24 hpi). Pyruvate increased for both pathovars, suggesting high bacterial respiration activity. As pyruvate acts as a link between glycolysis and TCA, it could be a target to explain Xt growth. The pathovars displayed

different metabolite profiles for sucrose, galactose, and amine pathways, except for stress-related metabolites (sorbitol, trehalose, spermidine, and galactinol), which accumulate later during infection, potentially indicating a bacterial or plant stress response (9, 18–20). The purine pathway includes metabolites involved in nucleic acids, amino acids, and sugar metabolism. Hypoxanthine, guanine, guanosine, and adenine accumulated, and hypoxanthine had the highest FC at 24 hpi. These metabolites are part of the purine salvage pathway to recover bases and nucleosides (21). These changes occurred later during infection suggesting metabolite recycling. The pentoses and glucuronate interconversions pathway only changed in Xtu-BMAF and involved metabolites associated with the plant cell wall (xylose, xylonolactone, and xylonate), potentially providing carbon to the bacteria or resulting from cell lysis. Finally, we did not observe distinct amino acid trends for each pathovar BMAF, but both accumulated asparagine (6 hpi), aspartate (12 hpi), and glutamine (12 hpi), serving as potential carbon sources.

## DISCUSSION

Our study showed metabolic patterns for mesophyll colonization. The results demonstrated that the host and time were the overall main drivers of metabolic responses (Fig. 1A). Individual metabolites were affected differently by each pathovar. Xtu is a broad-host-range pathogen, and we suspect that the broad metabolic signature aligns with Xtu's generalist behavior, using widely available resources, while Xtt is a specialist with limited metabolism (22). Xtt has a narrow host range and can colonize the plant xylem, a hypothesized nutrient-limited space made of primarily dead cells. This, therefore, aligns with the understanding that vascular, xylem pathogens like Xtt have limited metabolic flexibility as they primarily interact with non-living tissue, while pathogens like Xtu thrive in a dynamic, living mesophyll environment.

*Xanthomonas translucens* species can manipulate plant genes that modulate abscisic acid (ABA) production and sucrose transport to spread (23, 24). In our study, we did not find any differences in ABA accumulation or metabolites associated with the carotenoid or terpenoid pathways specific to Xtu. Instead, we observed an increase in sucrose during infection. Although pathovars like *X. translucens* pv. cerealis do not induce the expression of SWEET sucrose transporters in wheat (25), the sucrose accumulation in barley could be due to the activity of other transporters like SUT (sucrose–proton symporter). Additionally, sucrose accumulation serves as a signaling molecule for plant growth and stress responses, suggesting that further research, including barley gene expression analysis, can validate this hypothesis. Our study highlights differences in Xt-BMAF metabolic pathways and shared responses associated with stress, salvage pathways, and carbon and nitrogen relocation. Future research should focus on understanding the role of metabolites as energy sources or signals for colonization, pathogen manipulation of host metabolic responses, and metabolic plant responses associated with resistance.

## ACKNOWLEDGMENTS

Support was provided by the following sources: a joint National Science Foundation–NIFA PBI grant (2018-05040 to J.M. Jacobs); CFAES Environmental Fellowship and the Ohio State University Presidential Fellowship (to N. Heiden); and USDA Hatch OHO01470 and Foods for Health, a focus area of the Discovery Themes at the Ohio State University (to J. Cooperstone).

## AUTHOR AFFILIATIONS

[1]Department of Plant Pathology, The Ohio State University, Columbus, Ohio, USA
[2]Infectious Diseases Institute, The Ohio State University, Columbus, Ohio, USA
[3]Department of Horticulture and Crop Science, The Ohio State University, Columbus, Ohio, USA

[4]Department of Food Science and Technology, The Ohio State University, Columbus, Ohio, USA

## PRESENT ADDRESS

Veronica Roman-Reyna, Department of Plant Pathology and Environmental Microbiology, Pennsylvania State University, University Park, Pennsylvania, USA

## AUTHOR ORCIDs

Veronica Roman-Reyna  http://orcid.org/0000-0003-0072-8096
Nathaniel Heiden  http://orcid.org/0000-0001-7673-5693
Jessica L. Cooperstone  http://orcid.org/0000-0001-7920-0088
Jonathan M. Jacobs  http://orcid.org/0000-0002-1553-2013

## FUNDING

| Funder | Grant(s) | Author(s) |
|---|---|---|
| USDA's National Institute of Food and Agriculture | 2018-05040 | Jonathan M. Jacobs |
| OSU | College of Food, Agricultural, and Environmental Sciences, Ohio State University (CFAES) | | Nathaniel Heiden |
| U.S. Department of Agriculture (USDA) | OHO01470 | Jessica L. Cooperstone |

## AUTHOR CONTRIBUTIONS

Veronica Roman-Reyna, Conceptualization, Data curation, Formal analysis, Investigation, Methodology, Project administration, Resources, Software, Supervision, Validation, Visualization, Writing – original draft, Writing – review and editing | Nathaniel Heiden, Methodology, Validation, Writing – review and editing | Jules Butchacas, Methodology, Validation | Hannah Toth, Methodology, Validation | Jessica L. Cooperstone, Methodology, Software, Supervision, Writing – original draft, Writing – review and editing | Jonathan M. Jacobs, Conceptualization, Funding acquisition, Project administration, Resources, Supervision, Writing – original draft, Writing – review and editing

## DATA AVAILABILITY

Materials and Methods are available on protocols.io (https://protocols.io/view/extraction-and-analysis-of-primary-metabolites-dur-c8ayzsfw). Raw cdf files have been deposited to the EMBL-EBI MetaboLights database with the identifier MTBLS7676. The peak intensities and the fold change data were deposited in figshare (https://doi.org/10.6084/m9.figshare.25097108.v2).

## ADDITIONAL FILES

The following material is available online.

### Open Peer Review

**PEER REVIEW HISTORY (review-history.pdf).** An accounting of the reviewer comments and feedback.

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
