## [Reviewer comments · Microbiology Spectrum]

Microbiology Spectrum

The timing of bacterial mesophyll infection shapes the leaf chemical landscape.

Veronica Roman-Reyna, Nathaniel Heiden, Jules Butchacas, Hannah Toth, Jessica Cooperstone, and Jonathan Jacobs

Corresponding Author(s): Jonathan Jacobs, Ohio State University

Review Timeline:

Submission Date:	December 8, 2023
Editorial Decision:	January 2, 2024
Revision Received:	February 13, 2024
Accepted:	February 14, 2024

Editor: Lindsey Burbank

Reviewer(s): The reviewers have opted to remain anonymous.

Transaction Report:

DOI: <https://doi.org/10.1128/spectrum.04138-23>

Re: Spectrum04138-23 (The timing of bacterial mesophyll infection shapes the leaf chemical landscape.)

Dear Prof. Jonathan M Jacobs:

Thank you for the privilege of reviewing your work. Below you will find my comments, instructions from the Spectrum editorial office, and the reviewer comments. The study and associated data are important to understand the interactions of *Xanthomonas* pathogens in the plant apoplast. There are just a few suggestions on clarity and providing more details on the methodology.

Revision Guidelines

Sincerely,
Lindsey Burbank
Editor
Microbiology Spectrum

Reviewer #1 (Comments for the Author):

The manuscript "The timing of bacterial mesophyll infection shapes the leaf chemical landscape" by Roman-Reyna et al. characterize metabolic landscape of barley over time through colonization of the mesophyll space by two barley-infecting lineages, Xtu and Xtt. The main concern is related to the experimental procedures, where the details of the experiment are not clearly presented.

Major concerns:

1. Metabolomics experiments and derived information are needed in reviewing, while the raw data and the methodology could

not be found using the accession number provided. The raw data must be accessible at the time of review, and available to reviewers.

2.Line 70-75: The authors summarized the experimental procedures in one paragraph, while the details of the experiment are not clear. The authors should provide more details about their experiment, e.i. the age of plants which they have inoculated, the OD of bacterial suspension, the name of barley cultivar, as well as whether they have pooled treated leaves in one tube or not.

3.The authors should explain how they considered or analyzed the metabolites lacking full structural identification, and then how they normalized their data.

Minor comment:

Line 74: When using a term for the first time, we are not allowed to use abbreviations of that term. Please replace full name of GC-TOF MS term in the text.

Line 107: The authors reported that sucrose accumulates in the Xtu-treated leaves at 12hpi. As we know, the Xtu strain XT4699 is able to induce a host gene, encoding 9-cis-epoxycarotenoid dioxygenase (TaNCED-5BS) gene, which modulates abscisic acid levels and promotes spread of bacteria within infected leaves of wheat (Peng et al. 2019, PNAS), on other hand, Shah et al.'s 2023 study (Phytopathology) used qRT-PCR and RT-PCR to examine 41 TaSWEETs genes, but none of them revealed any differential expression in response to *X. translucens* pv. *cerealis* (Xtc) pathogen infection of wheat. Both studies show SWEET genes are not targeted by Xt pathogens. Now I just wonder how sucrose can accumulate in the plant when SWEET genes are not targeted.

Reviewer #2 (Comments for the Author):

This study describes metabolomic comparison of apoplastic fluid from barley plants infected with two different *Xanthomonas* pathogens. As an observation and description of a good dataset I think it is important and could lay the groundwork for a lot of future research to better understand these pathogenic bacteria and their interactions with the host. I know that it is submitted as a short form article and that there are some limitations associated with that, but there are a few things which would make it easier to follow and more useful to the research community. First, if a more detailed description of the methods could be provided that would help with replication and future research. Maybe a detailed protocol could be included as supplemental material or linked elsewhere (protocols.io, other published work etc). Second, the text is very broken up and jumps around a bit. Maybe use of subheadings or overall re-working some of the text could make the main points stand out more. For example the first couple paragraphs could be combined into a concise introduction and descriptions of the results could be consolidated under subheadings describing the main findings.

Minor corrections:

Line 91: should be Xtt-BMAF?

Line 97: accumulated in pathovar-infected samples?

The manuscript “The timing of bacterial mesophyll infection shapes the leaf chemical landscape” by Roman-Reyna et al. characterizes metabolic landscape of barley over time through colonization of the mesophyll space by two barley-infecting lineages, Xtu and Xtt. The main concern is related to the experimental procedures, where the details of the experiment are not clearly presented.

Major concerns:

1. Metabolomics experiments and derived information are needed in reviewing, while the raw data and the methodology could not be found using the accession number provided. The raw data must be accessible at the time of review, and available to reviewers.
2. Line 70-75: The authors summarized the experimental procedures in one paragraph, while the details of the experiment are not clear. The authors should provide more details about their experiment, *e.i.* the age of plants which they have inoculated, the OD of bacterial suspension, the name of barley cultivar, as well as whether they have pooled treated leaves in one tube or not.
3. The authors should explain how they considered or analyzed the metabolites lacking full structural identification, and then how they normalized their data.

Minor comment:

Line 74: When using a term for first time, we are not allowed to use abbreviations of that term. Please replace full name of GC-TOF MS term in the text.

Line 107: The authors reported that sucrose accumulates in the Xtu-treated leaves at 12hpi. As we know, the Xtu strain XT4699 is able to induce a host gene, encoding 9-cis-epoxycarotenoid dioxygenase (*TaNCED-5BS*) gene, which modulates abscisic acid levels and promotes spread of bacteria within infected leaves of wheat (Peng et al. 2019, PNAS), on other hand, Shah et al.'s

2023 study (Phytopathology) used qRT-PCR and RT-PCR to examine 41 *TaSWEETs* genes, but none of them revealed any differential expression in response to *X. translucens* pv. *cerealis* (Xtc) pathogen infection of wheat. Both studies show *SWEET* genes are not targeted by XT pathogens. Now I just wonder how sucrose can accumulate in the plant when *SWEET* genes are not targeted.

Response to reviewers

We thank the reviewers for the constructive feedback. We apologize for not having the data available during the review. We were unaware of the delay in making the data publicly available, the link to the study is public now in MetaboLights (<https://www.ebi.ac.uk/metabolights/editor/MTBLS7676>).

Please see the responses below.

Reviewer #1:

Major concerns:

1. Metabolomics experiments and derived information are needed in reviewing, while the raw data and the methodology could not be found using the accession number provided. The raw data must be accessible at the time of review, and available to reviewers.

2. Line 70-75: The authors summarized the experimental procedures in one paragraph, while the details of the experiment are not clear. The authors should provide more details about their experiment, e.i. the age of plants which they have inoculated, the OD of bacterial suspension, the name of barley cultivar, as well as whether they have pooled treated leaves in one tube or not.

3. The authors should explain how they considered or analyzed the metabolites lacking full structural identification, and then how they normalized their data.

This is for the three major concerns. The protocol and raw data are deposited in Metabolights, the link is now public. The tables to create the figures are in <https://doi.org/10.6084/m9.figshare.25097108.v2>, and a very detailed methods description is on protocols.io (<https://protocols.io/view/extraction-and-analysis-of-primary-metabolites-dur-c8phzvj6>).

Minor comment:

Line 74: When using a term for the first time, we are not allowed to use abbreviations of that term. Please replace full name of GC-TOF MS term in the text.

We added the full name.

Line 107: The authors reported that sucrose accumulates in the Xtu-treated leaves at 12hpi. As we know, the Xtu strain XT4699 is able to induce a host gene, encoding 9-cis-epoxycarotenoid dioxygenase (TaNCED-5BS) gene, which modulates abscisic acid levels and promotes spread of bacteria within infected leaves of wheat (Peng et al. 2019, PNAS), on other hand, Shah et al.'s 2023 study (Phytopathology) used qRT-PCR and RT-PCR to examine 41 TaSWEETs genes, but none of them revealed any differential expression in response to *X. translucens* pv. *cerealis* (Xtc) pathogen infection of wheat. Both studies show

SWEET genes are not targeted by Xt pathogens. Now I just wonder how sucrose can accumulate in the plant when SWEET genes are not targeted.

We thank the reviewer for bringing attention to the ABA and SWEET topics. We added part of the answer to the manuscript discussion.

In response to your inquiry regarding abscisic acid (ABA), we did not observe it among the annotated or differentially accumulated metabolites. We also investigated the carotenoid biosynthesis and terpenoid backbone biosynthesis as both are precursors pathway. Notably, in the terpenoid pathway, we observed the accumulation of 2-deoxyerythritol at 12 hours in both Xt and Xtu samples. It is worth mentioning that Xtu, primarily associated with wheat, was the focus of a study by Peng et al. (2019, PNAS). Our study focused on barley, and we did not test for changes in ABA for this study.

Plants have distinct sugar transporter families, including SWEETs (passive sugar uniporters) and sucrose transporters-SUT (sucrose-proton symporter) [1s]. This diversity suggests the possibility of sucrose accumulation through the activity of various transporters. Additionally, sucrose accumulation serves as a signaling molecule for plant growth and stress responses [2s]. Although validating this hypothesis in barley requires gene expression analysis, it is crucial to note that the primary focus of this paper was to generate metabolomic data, establishing a valuable resource for future investigations. For the same reasons, we were cautious in not explicitly addressing SWEET transporters in this context. Moreover, we suspect that water-soaking symptom development induced by ABA in wheat may increase intercellular water levels and indirectly trigger sucrose efflux in a SWEET independent manner that is independent of SWEET gene induction.

[1s] J. Chen, *et al.*, Starving the enemy: how plant and microbe compete for sugar on the border. *Frontiers in Plant Science* **14** (2023).

[2s] S. Kircher, P. Schopfer, Photosynthetic sucrose acts as cotyledon-derived long-distance signal to control root growth during early seedling development in Arabidopsis. *Proceedings of the National Academy of Sciences* **109**, 11217–11221 (2012).

Reviewer #2:

First, if a more detailed description of the methods could be provided that would help with replication and future research. Maybe a detailed protocol could be included as supplemental material or linked elsewhere (protocols.io, other published work etc).

We followed the advice. We uploaded all methods to protocol.io with the link <https://protocols.io/view/extraction-and-analysis-of-primary-metabolites-dur-c8ayzsfw> We added the link to data availability.

Second, the text is very broken up and jumps around a bit. Maybe use of subheadings or overall re-working some of the text could make the main points stand out more. For

example the first couple paragraphs could be combined into a concise introduction and descriptions of the results could be consolidated under subheadings describing the main findings.

We changed the structure of the document by adding subtitles and sections.

Minor corrections:

Line 91: should be Xtt-BMAF?

Changed to Xtt-BMAF.

Line 97: accumulated in pathovar-infected samples?

We changed to “.. both pathovar-infected samples..”

Re: Spectrum04138-23R1 (The timing of bacterial mesophyll infection shapes the leaf chemical landscape.)

Dear Prof. Jonathan M Jacobs:

Your manuscript has been accepted, and I am forwarding it to the ASM production staff for publication. Your paper will first be checked to make sure all elements meet the technical requirements. ASM staff will contact you if anything needs to be revised before copyediting and production can begin. Otherwise, you will be notified when your proofs are ready to be viewed.

Sincerely,
Lindsey Burbank
Editor
Microbiology Spectrum